# In Situ TEM Observation of Electric Field-Directed Self-Assembly of PbS and PbSe Nanoparticles

**DOI:** 10.3390/nano15161275

**Published:** 2025-08-18

**Authors:** Iryna Zelenina, Harald Böttner, Marcus Schmidt, Yuri Grin, Paul Simon

**Affiliations:** 1Max-Planck-Institut für Chemische Physik Fester Stoffe, Nöthnitzer Str. 40, 01187 Dresden, Germany; irynaszelenina@gmail.com (I.Z.); marcus.schmidt@cpfs.mpg.de (M.S.); juri.grin@cpfs.mpg.de (Y.G.); 2Fraunhofer Institut für Physikalische Messtechnik, Georges-Köhler-Allee 30, 79110 Freiburg, Germany

**Keywords:** lead selenide, lead sulfide, in situ TEM, nanoparticles, thin film, electric field, thermodiffusion

## Abstract

Nano-sized particles of semiconducting lead sulfide and selenide and their 2D thin layers show high potential in applications, such as field-effect transistors, photodetectors, solar cells, and thermoelectric devices. The generation of PbS and PbSe nanobars and nanocubes is evoked by in situ electron beam treatment, leading to the formation of thin, extended 2D nanolayers. The initial single crystals are decomposed via sublimation of PbS and PbSe in terms of molecular and atomic fragments, which finally condense on the cold substrate to form nanostructures. The fragments in the gas phase were proven using mass spectrometry. In the case of PbS, Pb^+^ and PbS^+^ species could were detected, whereas PbSe disintegrated into Pb^+^, Se_2_^+^, and PbSe^+^. The threshold current that initiates fragmentation increases from PbTe via PbSe up to PbS, which is in line with the increasing crystal formation energies. The uniform orientation of independently formed nanoparticles on the macroscopic scale can be explained by an external electric field acting on emerging dipolar nanospecies. The external dipole field originates from the sputtered mother crystal, where the electron flux is initiated; thus, a current arises between the crystal’s hot and cold ends. On the contrary, in small single crystals, due to the lack of sufficient charge carriers, only local material excavation is detected instead of extended depletion and subsequent nanoparticle deposition. This fragmentation process may represent a new preparation route that provides lead chalcogenide nanofilms that are free of contamination or surfactant participation, which are typical drawbacks associated with the application of wet chemical methods.

## 1. Introduction

Specific properties and potential applications in nanotechnology have placed nanosized materials in focus of substantial scientific interest. Nowadays, performance improvements in quickly developing fields, like solar energy panels, infrared lasers, window coatings, and energy generation, can be gained by using semiconductor nanoparticles [1,2]. The morphology and size of semiconductor nanoparticles defines their physical properties. Therefore, the best way to achieve improved working performance is by controlling these two parameters [3,4].

Among the large variety of semiconductor nanomaterials, lead chalcogenides have attracted attention because of their useful physical properties, low-cost production and a variety of synthesis routes making PbSe and PbS attractive for feasible applications [5,6,7]. The possibility of synthesizing nanoparticles of a certain shape (1D) and subsequently triggering oriented nanoparticle attachment that leads to nanofilm formation (2D or 3D) is a further advantage for controlling thickness and homogeneity [8,9]. In the case of chemically deposited films, the morphology of the films is dependent on the thickness. Few investigations have established the relationship between the growth of the grain size and the increase in thickness [10,11]. The lead chalcogenide band gap value is closely related to the grain size of the deposited film [12,13]. When using the thermal evaporation technique for PbSe thin film deposition, the band gap varies in the range of 1.42–1.62 eV [14]. On the other hand, high photosensitivity has been reported for PbS thin films with smaller-sized grains that contain structural defects. This finding shows that photoconductivity is very sensitive to the material’s microstructure [15,16,17].

Many attempts have been made to prepare lead chalcogenide nanofilms using chemical techniques, such as the sonochemical route from an aqueous solution of lead acetate and sodium selenosulfate in the presence of trisodium citrate as a complexing agent. Other methods involve a solution reaction of lead nitrate and sodium selenosulfate, or the reaction of lead acetate with thioacetamide in dimethylformamide as a solvent and oleic acid and tri-n-octylphosphine diphenylether as surfactants [18,19,20,21]. At the same time, physical techniques have received much less attention [22,23]. In the current study, we elaborate on a preparation method for PbSe and PbS nanofilms based on sublimation and self-assembly on the surface under converged electron beam impact in TEM. We compare these results with our former work involving in situ nanoparticle/nanofilm morphogenesis from bulk PbTe crystals [24].

## 2. Materials and Methods

### 2.1. Crystal Growth and Preparation of TEM Specimens

PbSe and PbS single crystals were synthesized by applying an unseeded vapor transport technique using the respective elements as starting materials [25].

### 2.2. TEM

The crystals of PbSe and PbS were investigated using a FEI Tecnai F30-G2 transmission electron microscope at an acceleration voltage of 300 kV. TEM micrographs were recorded on a slow-scan wide-angle CCD camera (MultiScan, 2 × 2 k pixels; Gatan Inc., Pleasanton, CA, USA). The point resolution of the TEM was 2.0 Å, and the information limit was around 1.2 Å.

High-resolution TEM (HR-TEM) of the PbSe single crystal was carried out using the double-corrected JEM-ARM300F electron microscope operating at 300 kV (Dresden Grand ARM, JEOL Company, Akishima, Japan). The spherical aberration (Cs) of both the condenser and objective lens was corrected using dodecapole correctors in the beam (STEM) and the image (TEM) formation system. HR-TEM images were captured using a 4k × 4k pixel CCD array (Gatan US4000, Gatan Inc., Pleasanton, CA, USA), providing a TEM resolution of 0.66 Å.

The central parts of the original crystals were used for structural investigations. The specimens were hand-ground in a mortar with minimum force. The obtained powders were mixed with methanol and transferred to a TEM grid coated with holey carbon.

### 2.3. Mass Spectrometry

The gaseous phase released during the thermal decomposition of PbS, PbSe, and PbTe was investigated using a thermogravimetry-coupled mass spectroscopy (TG-MS) system, combining a STA 409 CD thermobalance (NETZSCH, Selb, Germany) and a QMS 422 quadrupole mass spectrometer (Pfeiffer Vacuum GmbH, Asslar, Germany). The installation of this system in an argon-filled glove box (MBraun, Inertgas-Systeme GmbH, Garching, Germany) allowed the handling and measurement of samples that were particularly sensitive to air and/or moisture.

The individual samples were measured under the following conditions: temperature range of 25–1100 °C, sample mass of 12–17 mg, heating and cooling rates of 5 K/min, using a corundum crucible with a perforated lid and a type S (PtRh/Pt) thermocouple. The mass spectra were measured in the range of 1 < *m*/*z* < 500 amu using 70 eV electron impact ionization. The measurements were performed in a flowing argon atmosphere used as the purge gas (Ar 99.999% 50 mL/min), followed by drying and oxygen post-purification using a Big Oxygen Trap from Trigon Technologies (Rancho Cordova, CA, USA).

During the heat treatment, the gases evolving from the sample were mixed with a continuous stream of argon purge gas and directed into the ionization chamber of the mass spectrometer through a skimmer located directly above the bore of the sample crucible. Background mass spectra were recorded under the same conditions prior to the measurement. The individual gas species were identified on the basis of their isotopic patterns and the mass-to-charge ratios of the ions.

## 3. Results

The pristine single crystals of PbS and PbSe show dimensions of 4 × 5 mm^2^ (PbS) and 3 × 3 mm^2^ (PbSe), as shown in Figure 1a and Figure 2a, respectively. Light microscopy images of the PbS single crystal show the original mm-sized single crystal (Figure 1a,b), which was further ground to obtain μm-sized particles. As shown in the TEM image, the sample was highly nanostructured and interspersed with dislocations (Figure 1c), whereas the selected area electron diffraction (SAED) from the [001] zone reveals a regular lattice (Figure 1d). At higher magnification, dislocations with lengths of 5–200 nm become obvious, along with nanodomains about 10 nm in size (Figure 1e,f). Light microscopy images of a pristine PbSe single crystal is shown in Figure 2a,b. Appendix A shows the EDX spectrum where a ± 5% deviation is observed from the ideal 1:1 compostion. Overview TEM images reveal a high defect density consisting of domains measuring 2–5 nm in diameter (Figure 2c,d). In the field of view of (25 nm)^2^, more than seven different nanosized defects can be observed (Figure 2e). The corresponding FFT shows superstructure reflections along the *a* and *b* directions, along with reflections of additional crystallites (Figure 2f). At the same scale, an ideal structure is also occasionally observed (Figure 2g), as confirmed by the Fourier transform of the [001] zone (Figure 2h). Figure 3a displays a magnified section from the bottom right of Figure 2a, which highlights a chessboard-like structure with superstructure (or modulation), indicated by an approximate doubling along the *a* and *b* directions in Figure 3b. Even in apparent ideal regions (Figure 3c), additional reflections indicate a superstructure along the (120) direction, with doubling along the *a* direction (Figure 3d). The superstructure may be explained by interstitials, such as Pb atoms (self-doping) in the octahedral sites, along with additional gap formation between the layers due to lone pair (dangling bonds) repulsion (Figure 3e) [24].

The fragmentation of the PbSe microparticle during TEM occurred as the electron beam was progressively converged. Gradual evaporation of the single-crystal particle started after the current of the convergent beam reached a certain threshold value. The total amount of the evaporated material in this experiment was approximately 25% of the original particle (Figure 4a,b). This amount is much smaller than in the case of a PbTe microparticle, where the total amount is more than 50% [24]. The produced vapor moved towards colder areas of the carbon substrate and precipitated in the form of rectangular-shaped particles. The precipitation products spread on the carbon film and covered approximately an area of 25 µm in diameter around the initial particle (Figure 4b).

The single-crystal PbS microparticles underwent the same process of fragmentation under a convergent electron beam as observed in PbSe and PbTe (Figure 4c,d). Approximately 25% of the initial particle volume was sublimated by the beam treatment. The decomposition products covered approximately 15 µm in diameter of the adjacent area and formed a uniform PbS nanofilm around the initial particle (Figure 4d). The thickness of the PbS nanofilm was below 100 nm thickness because it appeared transparent under the electron beam. It consisted of randomly oriented grains of different sizes and shapes. With increasing distance from the initial particle, the homogeneity of the PbS film changed: the homogeneous film turned into dispersed interconnected particles, with further transformation into isolated particles at the periphery. The reason for this inhomogeneity was the difference in the building material supply from the gas phase during the fragmentation process.

The time-resolved evolution of the nanoparticles of PbS is shown in two movies in the Appendix A. In overview mode at the macroscopic scale, within a field of view of 3.5 μm, the generation of a nanofilm can be observed just beneath the single-crystal PbS particle (Appendix A: PbS_movie 1), where the beam current density was about 1800 e/nm^2^. The nucleation and formation of nanoparticles, followed by their subsequent growth, is visualized at the nanoscale, within a field of view of 500 nm (Appendix A: PbS_movie 2). The applied electron dose was about 5000 e/nm^2^.

In the case of PbSe, similar behavior to PbS was observed. A large PbSe crystal was irradiated, and the dose was increased continuously. After 1.5 min of irradiation and exceeding a threshold value of 150 e/nm^2^, spontaneous sublimation and nanoparticle deposition was detected; see Appendix A: PbSe_movie 1. Another experiment showed a threshold value for start of the Seebeck effect at a critical dose of about 830 e/nm^2^, which is 5 times higher than for the previous experiment; see Appendix A: PbSe_movie 2. The dynamics of the thin film formed by the nanoparticles was also investigated (Appendix A: PbSe_movie 3). Here, a continuous elimination of smaller domains through their integration into larger aggregates was observed, which is known as Ostwald ripening (Figure 5 and Figure 6).

A thin film formed from fused nanoparticles around the irradiated PbSe crystal (red arrow, Figure 5a). A closer look reveals that the PbSe nanofilm consisted of oriented grains (Figure 5b). A large fraction of the observed grains was oriented close to the [100] zone along the viewing direction. The contrast in the micrograph (Figure 5b) indicates that the grains are not oriented in the same way. Darker contrast in the TEM images corresponds to grains oriented closely to the main zones where electrons are strongly diffracted; see, e.g., the grain marked A. Other grains are slightly off the {001} axis and appear brighter, as depicted by grain B. Grains with a large number of neighboring grains experience mechanical stress, as marked in the central grain denoted as C, where strong parallel lines indicate a wrinkling process. The difference in contrast between two grains indicates that they are twisted, and their border contains a screw dislocation (red arrow, Figure 5b,c) [24]. Fast Fourier transform of the vicinal grains in Figure 5c (inset) indicates a {001} orientation. Some grain borders have isolated dislocations that appear at certain intervals. This type of dislocation appears between two grains with a low attachment angle (red arrows, Figure 5d).

The same film formation mechanism was observed in the case of PbS, where PbS nanoparticles grew and merged until they filled all free areas near the initial particle (Figure 5e). Further away from the center, the film density diminished to such an extent that only interconnected nanoparticles were present (Figure 5f,g). The shapes of these particles varied and rarely resembled bars. Occasionally, isolated PbS particles showing perfect rectangular shape were present, but this shape type occurred infrequently (Figure 5h).

The homogeneity of the newly precipitated nanofilm varied with distance from the source particle. The adjacent carbon film around the source particle was covered with a PbSe nanofilm made of grains (Figure 6). Because the film appeared transparent under the electron beam, the film thickness was assumed to be less than 100 nm. With increasing distance from the source particle, the nanofilm density deteriorated, and the dense film steadily turned into interconnected nanoparticles. The particle size and distribution decreased in the areas further away from the initial particle. The reason for such structural variations is the same as in the case of PbTe: the decreasing supply of Pb, Se, and S atoms and PbS and PbSe molecules from the gas phase with increasing distance from the source crystal [24]. The process of nano-film formation in the proximity of the initial particle is straightforward. The supply of Pb and Se from the vapor precipitated and caused a high concentration of PbSe seeds formed per unit area. The excessive density of newly formed seeds accompanied by swift nanoparticle growth resulted in nanoparticles merging and the formation of a nanofilm (Figure 6a). Typically, when two nanoparticles come into contact, the larger particle absorbs the smaller one. However, some attachments were energetically stable, allowing both nanoparticles to continue growing without absorbing each other. Even after nanoparticles merged and formed a continuous nanofilm composed of grains, moderate changes in the structure still occurred. Larger grains gradually absorbed smaller ones, resulting in constant grain boundary changes (Figure 6a–d). A central grain (marked by a red circle) continuously grew in size and changed its shape from hexagonal (a) to octagonal (c,b) and, finally, to an irregular form (d). During its growth process, it absorbed material from neighboring grains (yellow arrow in Figure 6).

At the periphery, the density of the PbSe nanofilm degraded, developing into interconnected or isolated nanoparticles. The isolated nanoparticles formed shapes resembling rectangles or squares with truncated vertexes (Figure 7a), as a result of the slow growth of the {111} facets [26,27]. At the same time, less frequently, nanoparticles with completed rectangular shapes appeared (Figure 7b). The number of isolated particles was highest at distances larger than 10–15 μm from the initial particle. The area between the dense nanofilm and the isolated nanoparticles was filled with interconnected fragments. The different attachment angles resulted in various types of dislocations at the borders. Some adjacent particles were twisted relative to each other and had screw dislocations at their boundaries. This type of attachment is easily recognizable because adjacent particles show different contrast in images because of their different crystal orientations (diffraction contrast) (Figure 7c). When attached particles do not have a contrast difference and have the same orientation along the viewing direction, their boundary encompasses an edge dislocation (red arrows in Figure 7d).

The first step of PbS film formation involves abundant nucleation of seeds (Figure 7e,f). These seeds nucleate at various times, indicating that their nucleation is stochastic. Newly formed particles exhibit shapes that resemble rectangular bars as a result of free-energy minimization. During their growth, the shapes of PbS particles evolve differently. Some particles grow evenly in all directions and somewhat maintain a square form. On the other hand, particles frequently appear to grow more quickly in one of the main directions ([100], [010], or [001]) and result in elongated bars. The nanoparticles grow swiftly and unavoidably come into contact with each other as a result of the high density of the nucleated seeds per area.

When two PbS nanoparticles come into contact, the smaller particle is absorbed by the larger one in order to minimize the surface free energy. After the smaller particle merges with the main body of the larger particle, the resulting shape often deviates significantly from a rectangular form (red circles, Figure 7g,h). At the same time, some of the particles do not merge after coming into contact; instead, they form a dislocation at their boundary. The presence of dislocation is commonly energetically favorable because it requires less energy compared to merging.

The high-resolution TEM images of the nanocrystals generated by irradiation of single-crystal PbS mainly show particles oriented in the {001} direction (Figure 8a). In their initial state, they vary between 3 and 10 nm in size but can grow up to 60 nm or more. In several regions approximately 100 nm size, the nanoparticles displayed similar orientations, as indicated by the red arrows in Figure 8b,c. Sometimes the corners and edges of the cubic or rectangular nanoparticles appear incomplete, resulting in a truncated shape along the {110} direction (Figure 8d).

The common orientation of nanoparticles is likely promoted by atomic-sized links to their neighboring particles (Figure 9a). These interconnections between neighboring particles were observed on all facets of the crystals (Figure 9b). The central 20 nm-sized nanoparticle had seven contact points with its neighboring particles, with some of them marked by red arrows. Another example involving a smaller nanoparticle reveals an atomic-sized single-chain PbS interconnection, bridging a distance of about 5 nm, as indicated by the yellow arrow in Figure 9c. Again, fibrillar contacts to all neighboring particles were observed (Figure 9d, red arrows and yellow regions). Another atomically thin single-chain PbS is shown in Figure 10a, bridging a 2 nm gap between the corners of neighboring particles. Further connections are indicated by red arrows. The atomically thin PbS chain is connected to a bundle of disordered chains on the left, as shown in Figure 10b. These atomically thin PbS chains start to evolve, e.g., from the {110} facet at the corners, along the {100} direction (Figure 10c,d red arrows).

A preferred orientation of the nanoparticles was observed in the case of PbSe across a large region of interest (Figure 11). Figure 11a displays irradiated crystal surrounded by nanoparticles. Further irradiation leads to massive volume decrease of the initial particle (Figure 11b). In Figure 11c, an assembly of nanoparticles is shown taken from red rectangle area in Figure 11a. The FFT suggests a nematic order within a field of view of about 200 × 200 nm^2^, encompassing more than 400 individuals (Figure 11d). Further zoom (Figure 11e) taken from the right bottom of Figure 11c (red rectangle) shows about 100 nanoparticles with a narrow orientation distribution of about ±10° as indicated by the FFT (Figure 11f).

## 4. Discussion

### 4.1. Stability of Lead Chalcogenide Compounds and Mass Spectrometry

The lead chalcogenides PbTe, PbSe, and PbS demonstrated similar behaviors under convergent electron beam irradiation. They decomposed via sublimation through layer-by-layer removal from the surface, producing a vapor phase in the vicinity of the pristine particle. The elements in the vapor phase condensed in the colder areas on the supporting carbon film of the TEM grid and formed new lead chalcogenide nanostructures. The atomic and molecular species of the gas phase were confirmed using mass spectroscopy, as shown in Table 1 and Figure 12.

Heavier molecular fragments, such as Pb_2_Te_2_, likely also exist in the gas phase. However, these species lie beyond the detection limit of about 512 *m*/*z* of the mass spectrometer used. The detected gas species in our mass spectroscopy experiment have also previously been detected in earlier investigations on these compounds; see [24,28]. The in situ nanoparticle morphogenesis from bulk PbTe crystals through sublimation using electron beam irradiation was comprehensively investigated in a previous study [24]. The threshold current that initiates fragmentation varies among PbTe, PbSe, and PbS. PbSe and PbS particles require a higher electron beam current density than PbTe to initiate fragmentation [25]. Another notable difference between PbTe, PbSe, and PbS is the change in particle volume. PbTe particles lose more than 50% of their initial volume after fragmentation [25] while, in the case of PbSe and PbS, the particles lose approximately 25% of their initial volume (Figure 4). The difference in sublimation energies is also reflected in the different sublimation temperatures, e.g., for lead fragments (Pb^+^) for the lead chalcogenides, as measured using mass spectrometry in an argon atmosphere. The temperature for the initiation of thermal decomposition for PbS and PbSe were almost the same, starting at 970 ± 5 °C and 960 ± 5 °C, respectively. In the case of PbTe, thermal decomposition began at a lower temperature of 925 ± 5 °C (Table 2, Appendix A).

The observed thermal decomposition behavior is in accordance with the known thermodynamic stability data (Appendix A). This indicates that the sample was heated to these temperatures by the electrons in the TEM. Mass spectroscopy at medium vacuum was also carried out in the case of PbSe to evaluate the decomposition under vacuum conditions, which is prevalent in the TEM. The decomposition temperature significantly decreased by about 300 °C, from about 960 °C in argon atmosphere to about 660 °C under a vacuum of 2.8 × 10^−3^ mbar, as shown in Appendix A. The highest mass loss occurred at more elevated temperatures, as indicated by the inflection points of the experimental curves at 800 °C under vacuum and 1100 °C in argon.

The stability sequence under electron beam irradiation can be explained by the experimentally determined enthalpies for formation of the compounds of interest at 298 K (Appendix A) [29]. The formation energies for PbS and PbSe are almost the same, at about −100 kJ/mol, whereas PbTe shows a significantly lower value of about −70 kJ/mol. This difference was also observed during the electron irradiation experiments, where the PbTe proved to be much easier to decompose under electron beam irradiation than PbS and PbSe.

According to the calculated enthalpy of formation values at 0 K for lead chalcogenides (Appendix A) [30], PbS (with about −0.8 eV/atom) is significantly more stable than PbSe or PbTe (with about 0.5 eV/atom). However, PbS is far less stable than PbO, which has an enthalpy of formation of about −1.5 eV, which is twice as large as that for PbS. The enthalpy of formation between PbS and PbSe differs significantly, by about 40%. An extreme difference was found between PbO and PbTe, with PbO exhibiting a formation enthalpy about 300% higher than that of PbTe. Only the species with the highest symmetry—*Fm-*3*m* for PbS, PbSe and PbTe, and *P*4/*nmm* for PbO—were considered for discussion.

To assess and understand compound stability, one can also consider the lattice energies [31]. As expected, in the sequence of lead chalcogenides, PbO is the most stable compound (with a lattice energy of 3565 kJ/mol) followed by PbS, PbSe and, finally, PbTe, which shows the lowest lattice energy of about 2647 kJ/mol (Appendix A). Compared with PbTe, the lattice energy of PbS is only 10% higher, and that of PbSe is about 4% lower. Thus, lattice energies show similar trend to the formation enthalpy values. These observations are in line with the known crystal binding energies of lead chalcogenides, with energies of 32.6 eV/molecule for PbS, 32.5 eV/molecule for PbSe, and 31.4 eV/molecule for PbTe [32], as shown in Appendix A. The ionicity of the bonding was also assessed, and it was found that the ionic contribution decreases from PbS to PbTe.

The stability energies of lead chalcogenides based on cohesive energy and the heat of sublimation were discussed in reference [33]. The increasing heat of sublimation values were comfirmed in the sequence starting with PbTe 219 kJ/mol, followed by PbSe 227 kJ/mol, and ending up with PbS 235 kJ/mol [Appendix A]. However, it was emphasized that the binding energies per atom were significantly higher than those of molecules, which must be considered when comparing atomic depletion mechanisms with the molecular sublimation of PbSe or PbS fragments.

The similar behavior of lead chalcogenides under a convergent electron beam indicates that the driving force behind specimen fragmentation is the same for all. All three compounds exhibit inherited imperfections in their crystal structures, such as dislocations and precipitates, which may have influenced the behavior of the samples under the convergent beam. Such dislocations are common and can dominate even physical properties, as observed in bulk PbTe. In a recent study, these dislocations were found to be temperature-dependent and responsible for the semiconductor-to-metal transition in this compound [34]. In the case of PbTe, numerous other defects, such as nanopatterning, intergrowth, and vacancies, also play a decisive role in depletion behavior by further lowering the sublimation energy [24,34]. Various defects, such as edge and 45° dislocations, were also reported for PbS, though in mechanically pretreated or heated samples [35]. For PbSe, a high density of dislocations (4 × 10^11^/cm^2^) has been reported [36]. They exhibit ellipsoidal shape morphology and are 10–20 nm in length. The slightly distorted lattice is believed to result from interstitial defects due to the accommodation of highly diffused Se in the ordered crystal at high temperature (1223 K) and high vapor pressure during synthesis. Consequently, the loss of Se causes non-stoichiometric defect regions that induce lattice distortion and strain in PbSe single crystals.

### 4.2. Possible Electric Field Effects

Careful analysis of the TEM results leads to the conclusion that the in situ observation of the depletion process in lead chalcogenides single crystals and the morphogenesis and self-organization of nanoparticles is not a simple sputtering event, but is also accompanied by the occurrence of an intrinsic electric field. This effect has been previously observed and described by Wu et al. for the thermoelectric material AgPb_m_SbTe_m+2_ [37]. They proposed the presence of an electric dipole field induced within the irradiated crystal. This explains the synchronized orientation of independent nanoparticles formed by sublimation of the original crystal within an ROI of several hundred nm (Figure 8 and Figure 11) due to an external electric field acting on the emerging dipolar nanoparticles (Figure 13). The dipole structure of PbTe, for example, is described as the appearance of local structural dipoles that develop from an undistorted ground state upon warming [38]. This behavior is consistent with a simple thermodynamic model in which the emerging dipoles are stabilized in the disordered state at high temperature due to the extra configurational entropy, despite the fact that the undistorted structure has lower internal energy [38].

The process of nanoparticle morphogenesis is highly dynamic, encompassing particle movement, changes in tilting angles relative to the substrate, and shape changes during growth. During time-resolved experiments (see Appendix A and Figure 6 and Figure 7), the orientation of the particles—such as nanorods or sheets—changes. This may be attributed to a specific interaction between the negatively charged carbon substrate, due to electron irradiation, and the dipole moment of the nanoparticles. During their growth process, the particles may develop a strong electrostatic repulsion from the substrate as their dipole strength increases with size. This may lead to an orientational rearrangement of the nanoparticle, where the positively charged end attaches to the negatively charged substrate (Figure 14).

Future investigations could make use of in situ TEM holders with electric contacts where a focused ion beam (FIB), several microns in size and cut from a single crystal, is placed. In this way, the electric current/potential between the cold and hot ends of the sample could be measured using the four-point contact method, which is induced by electron irradiation. Further experiments could also encompass direct imaging of the evolution dipole stray fields using time-resolved electron holography or 4D STEM. In these experiments, the strength of the electrostatic potential can be measured and imaged.

## 5. Conclusions

The morphology of the fragmentation products showed the same features for all three lead chalcogenides. It started as a homogeneous nanofilm composed of grains near the initial particle and deteriorated into isolated nanoparticles towards the periphery. After fragmentation, the observed shapes of the peripheral particles deviated for different compounds. PbTe nanoparticles present finished rectangular shapes, whereas unfinished rectangles are seldomly present [24]. PbSe nanoparticles commonly form truncated rectangles and rarely show finished rectangular-shaped particles, which is likely due to diffusion problems. Meanwhile, PbS nanoparticles consistently present indefinite shapes. The interconnected nanoparticles of lead chalcogenides usually result in the presence of edge and screw dislocations. These types of dislocations were observed for all compounds. Sublimation of the specimens occurred through sequential layer-by-layer removal and led to the formation of lead chalcogenide nanofilms in the vicinity of the initial particle. The decomposition temperature was about 670 °C for PbSe under vacuum, as determined by electron bombardment in the TEM and confirmed using mass spectroscopy. The synchronous orientation of emergent nanoparticles around the irradiated initial crystal indicated the presence of an intrinsic dipole field. The nanoparticles were interconnected by atomic bridges which stabilize the orientational order within the nanoparticle assembly. This beam-induced fragmentation process is a new synthesis method that provides lead chalcogenide nanofilms which are free of contamination.

## Figures and Tables

**Figure 1 nanomaterials-15-01275-f001:**
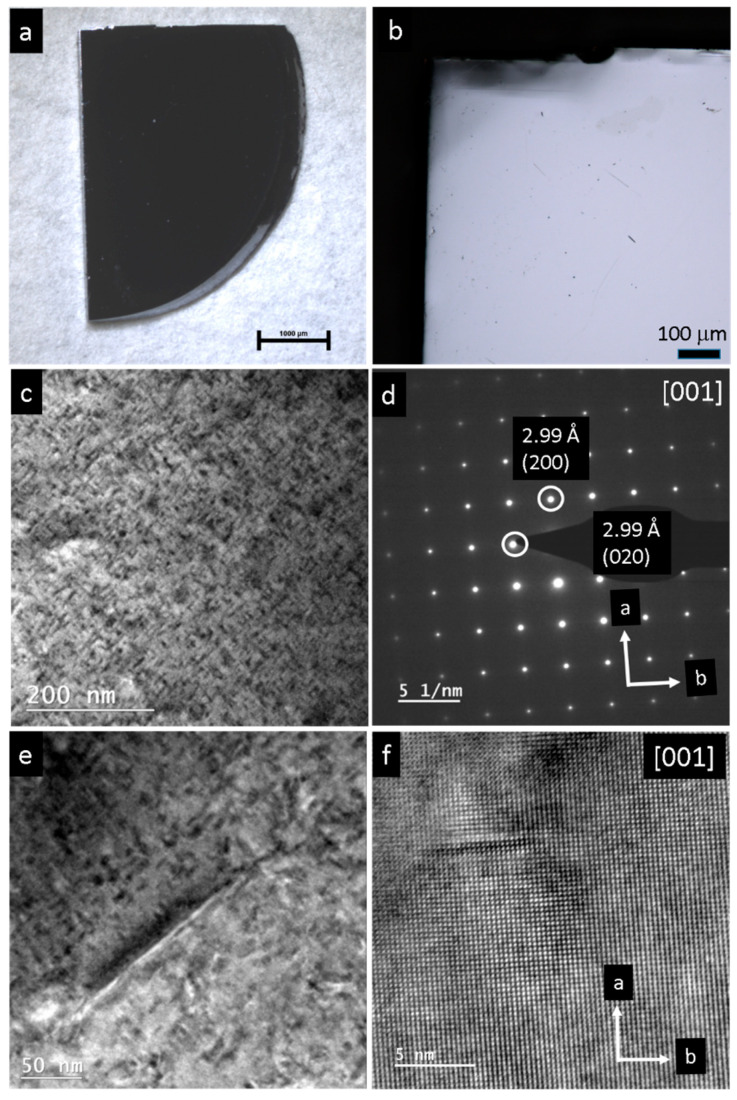
Pristine PbS single crystal: (**a**,**b**) Light microscopy image. (**c**) TEM overview image reveals a highly defect pattern consisting of nanodomain structuring and dislocations. (**d**) The electron diffraction shows a perfect single crystal pattern in the [001] orientation. (**e**) Magnification of (**c**) with a 200 nm-long dislocation in the center and nanodomains in the range of 10 nm in diameter. (**f**) High-resolution of PbS with regular lattice at bottom part and a dislocation at top left.

**Figure 2 nanomaterials-15-01275-f002:**
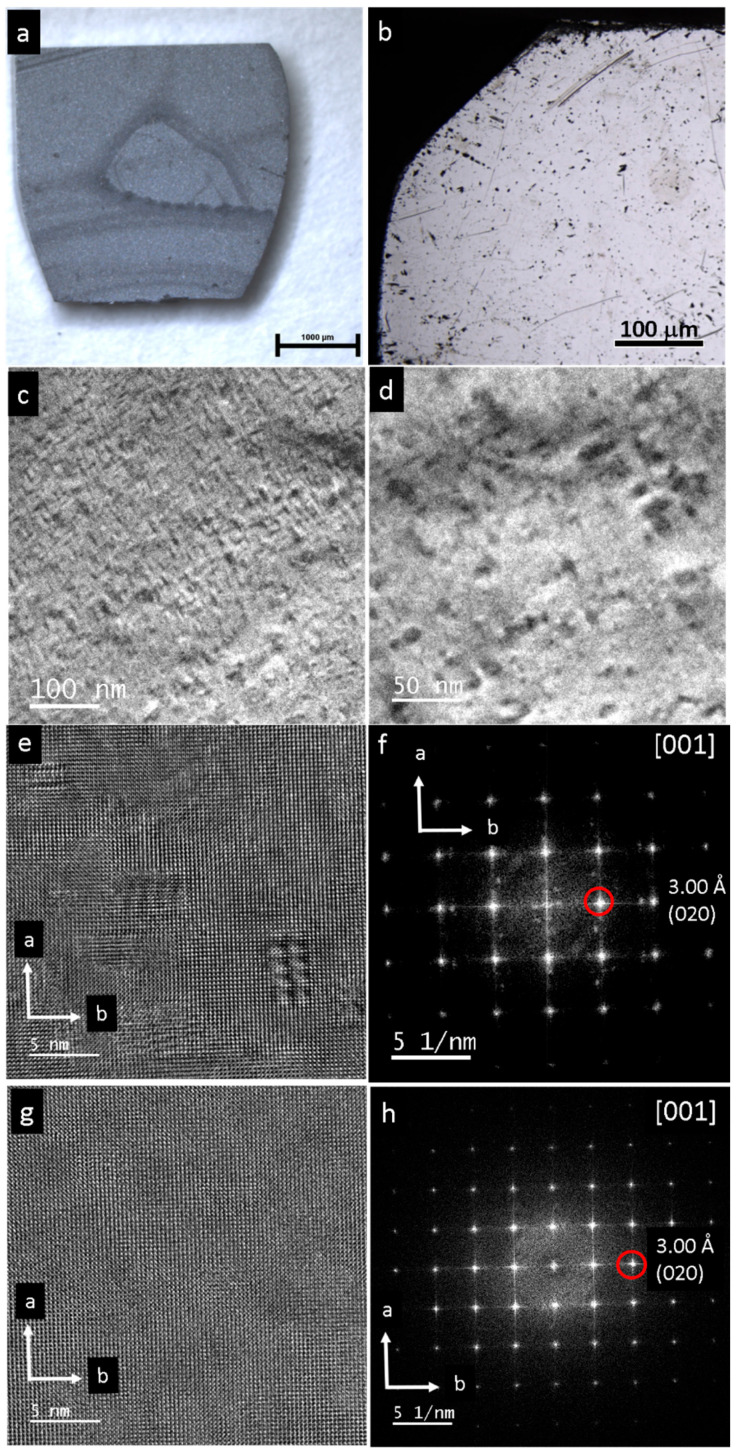
Pristine PbSe single crystal. (**a**,**b**) Light microscopy image. (**c**,**d**) TEM overview shows high density of dislocations and defects. (**e**,**f**) High-resolution and processed FFT in the [001] orientation. Occurrence of nanodomains in the range of 2–5 nm in diameter with additional reflections. (**g**,**h**) High-resolution image and derived FFT of PbSe with regular lattice.

**Figure 3 nanomaterials-15-01275-f003:**
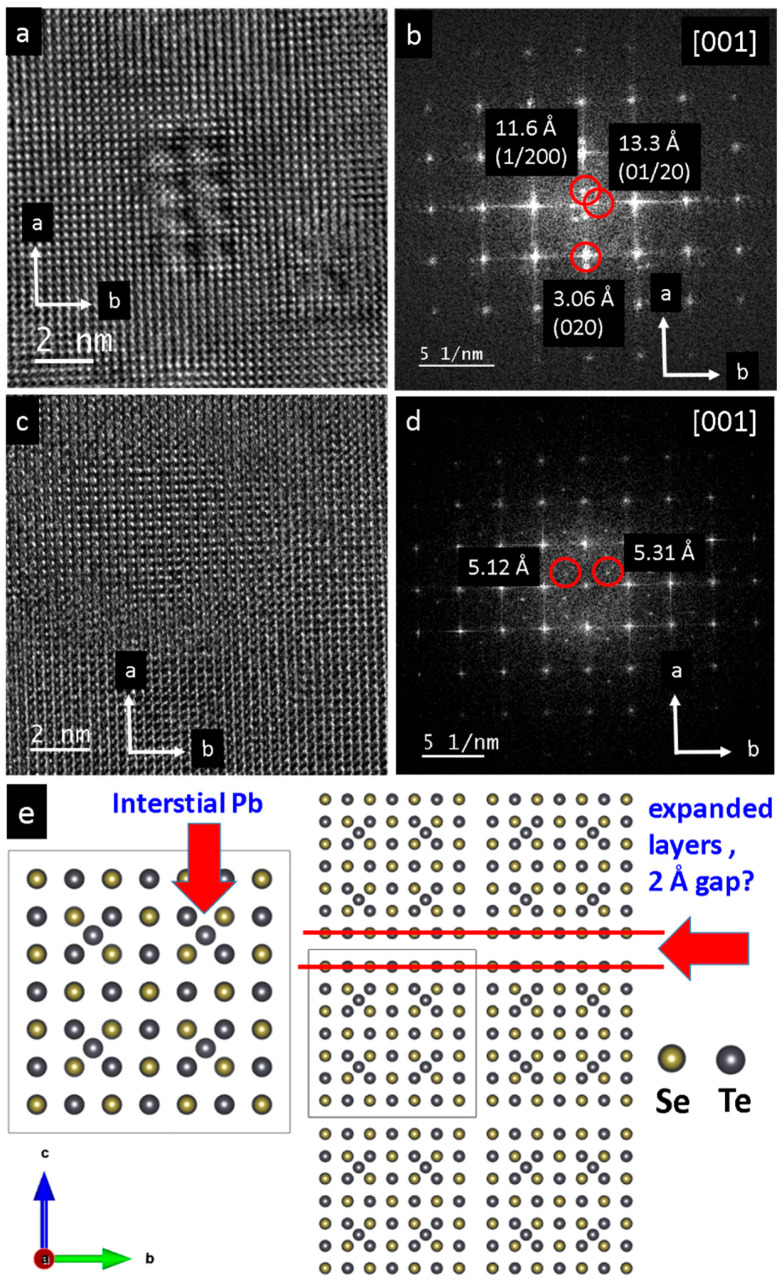
High-resolution TEM of the pristine PbSe single crystal. (**a**) Magnified area of Figure 2e, bottom right. The nanodomain consists of 8 units with a cell, which shows doubling of the regular unit cell along the *a* and *b* axes. (**b**) Processed FFT with additional reflections (red circles) indicating approximate doubling of the cell parameters. (**c**,**d**) High-resolution image and its derived FFT of pristine PbSe with apparent regular lattice. However, reflections at about 5.12 Å and 5.31 Å are also detected (red circles). (**e**) Possible model of chessboard structure imaged in (**a**); however, with 3-fold “superstructure” instead of doubling. Pb interstitials and gap formation due to lone pairs of Pb are indicated. Lead is indicated by grey; Se is indicated by yellow.

**Figure 4 nanomaterials-15-01275-f004:**
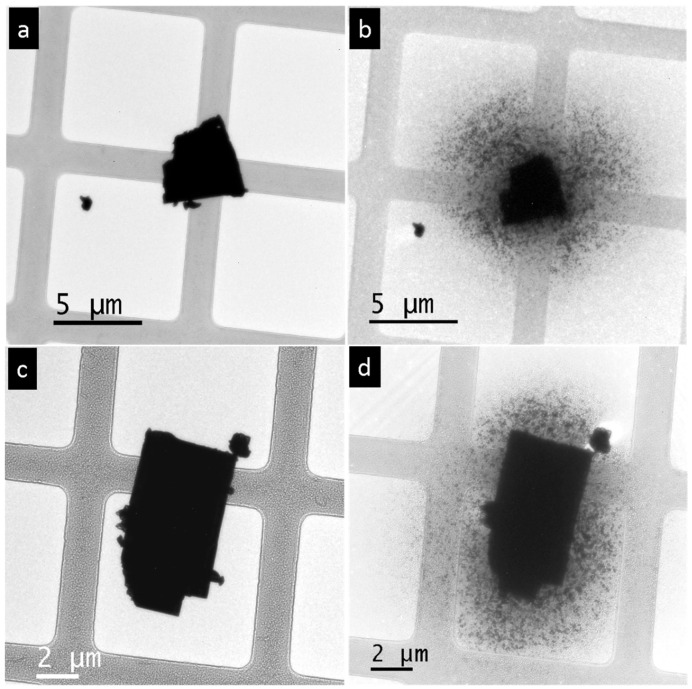
Convergent electron beam treatment of PbSe and PbS single crystals: (**a**) Original single-crystalline PbSe particle before heavy electron beam irradiation. (**b**) The same single-crystalline PbSe particle after irradiation is surrounded by fragmentation products. (**c**) Original single-crystalline PbS particle before electron beam irradiation. (**d**) The same PbS single-crystalline particle after irradiation, surrounded by fragmentation products.

**Figure 5 nanomaterials-15-01275-f005:**
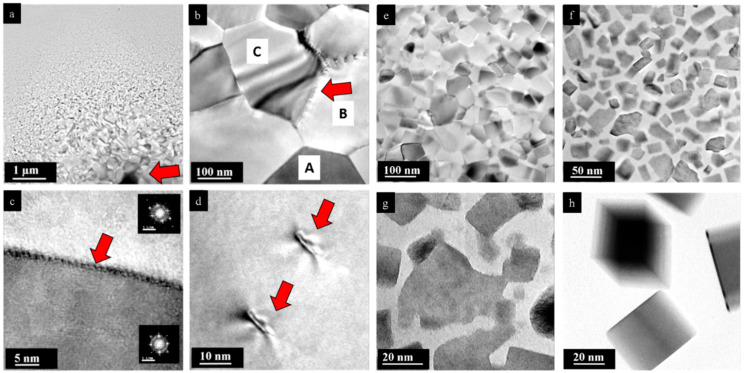
(**a**–**d**) PbSe nanofilm in the vicinity of the initial particle (indicated by red arrow): (**a**) PbSe nanofilm overview shows changes in morphology with increasing distance from the source particle. (**b**) Grains of PbSe forming different types of dislocations at their boundaries. (**c**) Screw dislocation at the boundary of two grains with orientation close to the [100] crystallographic direction. (**d**) Low-angle grain boundary of PbSe. (**e**–**h**) Nanostructured PbS. (**e**) Dense PbS nanofilm in the proximity of the initial particle. (**f**) Overview of interconnected nanoparticles. (**g**) Fused nanoparticles with non-regular shapes. (**h**) Separated individual PbS particles with regular rectangular shape.

**Figure 6 nanomaterials-15-01275-f006:**
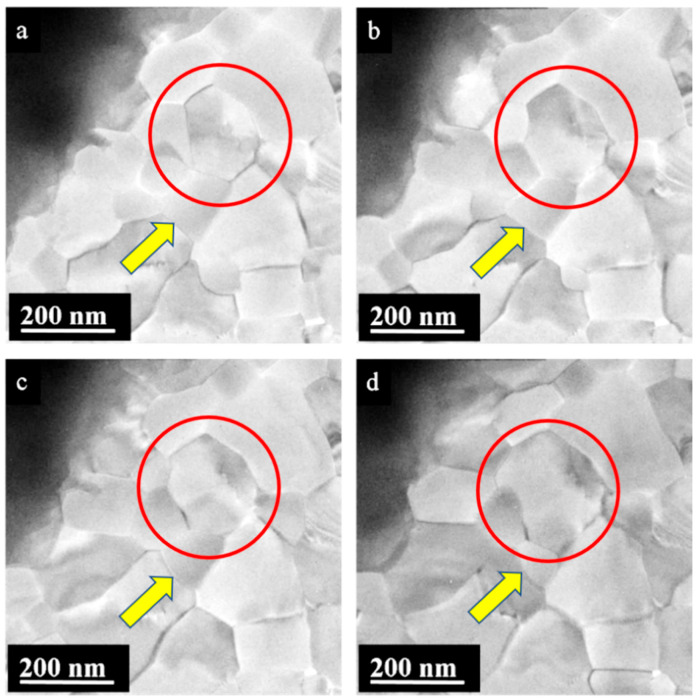
Dynamics (time-dependent series) of PbSe grain transformation during electron beam irradiation in the nanofilm. Some grains grew steadily (see red circle), whereas others continuously decreased in size and were resorbed (yellow arrow). (**a**) The initially formed PbSe nanofilm was composed of grains of different sizes and shapes. (**b**,**c**) Some of the smaller grains were gradually absorbed by larger grains. (**d**) Stabilized PbSe nanofilm with stable grain borders.

**Figure 7 nanomaterials-15-01275-f007:**
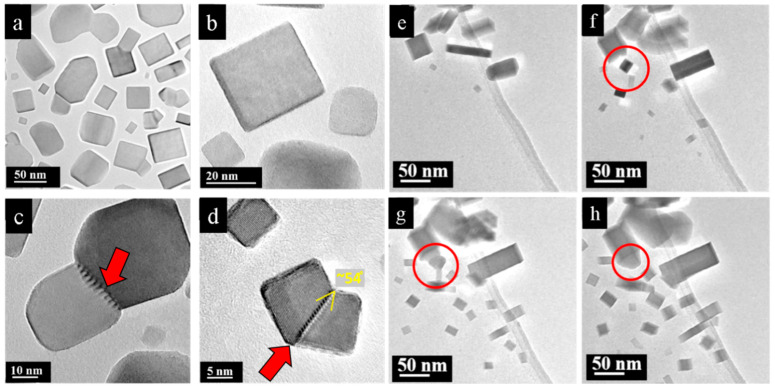
(**a**–**d**) Condensate products of single-crystal PbSe fragmentation after electron beam irradiation: (**a**) PbSe nanoparticles of different shapes (often truncated bars). (**b**) Perfect rectangle PbSe nanoparticle without truncated vertices. (**c**) Fused truncated squares with screw dislocation at their boundary (red arrow). (**d**) Merged nanoparticles with edge dislocation at their boundary (red arrow). (**e**–**h**) Formation and growth of PbS nanoparticles under electron beam irradiation, time series: (**e**,**f**) Stochastic nucleation of PbS seeds with further gradual growth on the carbon film while the initial micro-sized particle gradually evaporates. (**g**,**h**) Bars that come into contact merge, resulting in particles with shapes that significantly deviate from rectangular (red circles).

**Figure 8 nanomaterials-15-01275-f008:**
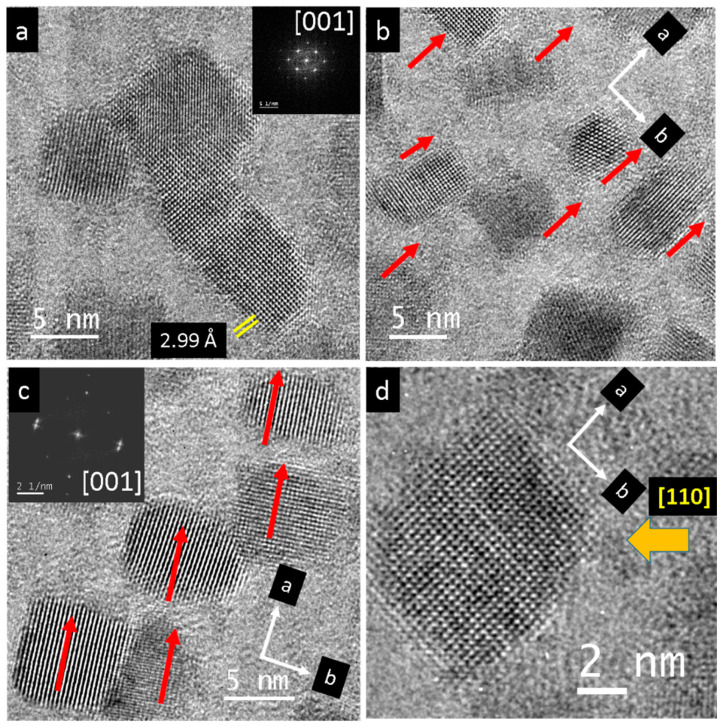
High-resolution images of common orientation of PbS nanocrystals produced by irradiation of a single crystal. (**a**) Particles were between 3 and 10 nm, mostly in the {001} orientation, cf. inset. (**b**,**c**) In certain regions, the nanoparticles display similar orientation, as indicated by red arrows. (**d**) Often, the edges were not complete, resulting in a truncated cube appearance.

**Figure 9 nanomaterials-15-01275-f009:**
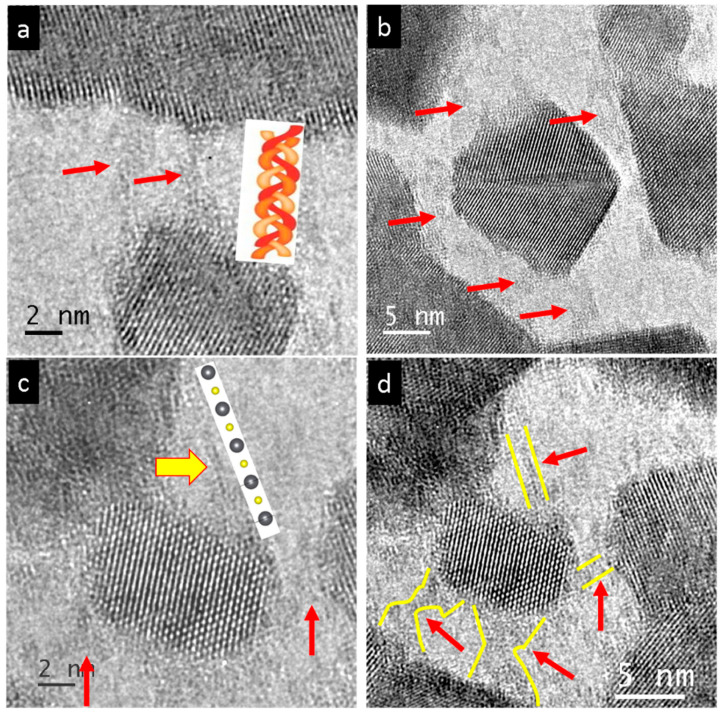
High-resolution images of PbS nanocrystalline products and their interconnection. (**a**) Bridges between neighboring particles (red arrows). (**b**) Central nanoparticle (size 20 nm) and its interconnections (red arrows) to neighboring particles. (**c**) Small nanoparticles show contacts where even a single-chain PbS serves as a bridging agent (yellow arrow). (**d**) Overview of the same particles and their contacts (red arrows and yellow regions).

**Figure 10 nanomaterials-15-01275-f010:**
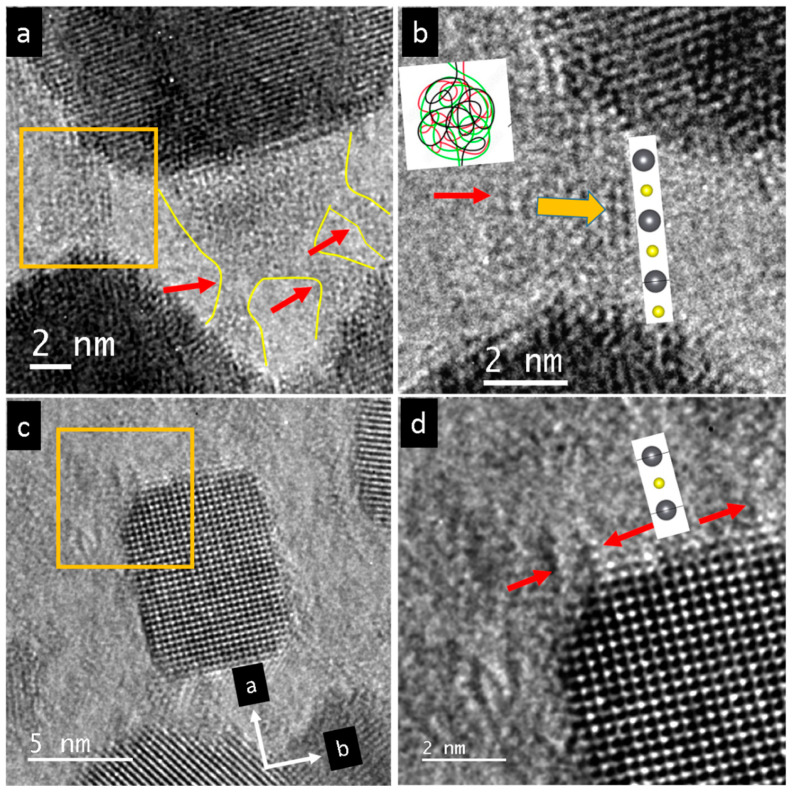
High-resolution images of PbS nanocrystal interconnections. (**a**) Starting from the corners of the cube’s (110) truncation surface, a single-chain PbS bridges to the neighboring particle. (**b**) Atomically thin PbS chain (orange arrow) connected to a bundle of disordered chains on the left (red arrow). (**c**,**d**) Single atomic PbS chains (see red arrows) evolved starting from (110).

**Figure 11 nanomaterials-15-01275-f011:**
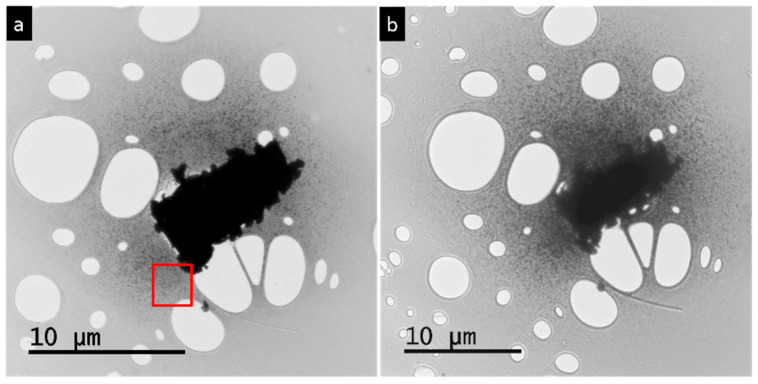
Preferred orientation of PbSe nanoparticles. (**a**) Macroscopic crystal (10 μm) after irradiation. (**b**) Irradiation led to extended deposition of nanoparticles within a circle (radius of 20 μm). (**c**) Assembly of about four hundred PbSe nanoparticles taken from red rectangle region marked in (**a**). (**d**) FFT revealed nematic order within a field of view of 200 × 200 nm^2^. (**e**) Magnification of (**c**), see red rectangle area in (**c**) at right bottom, field of view of 100 × 100 nm^2^, with about 100 particles. (**f**) FFT of (**e**) shows preferred orientation.

**Figure 12 nanomaterials-15-01275-f012:**
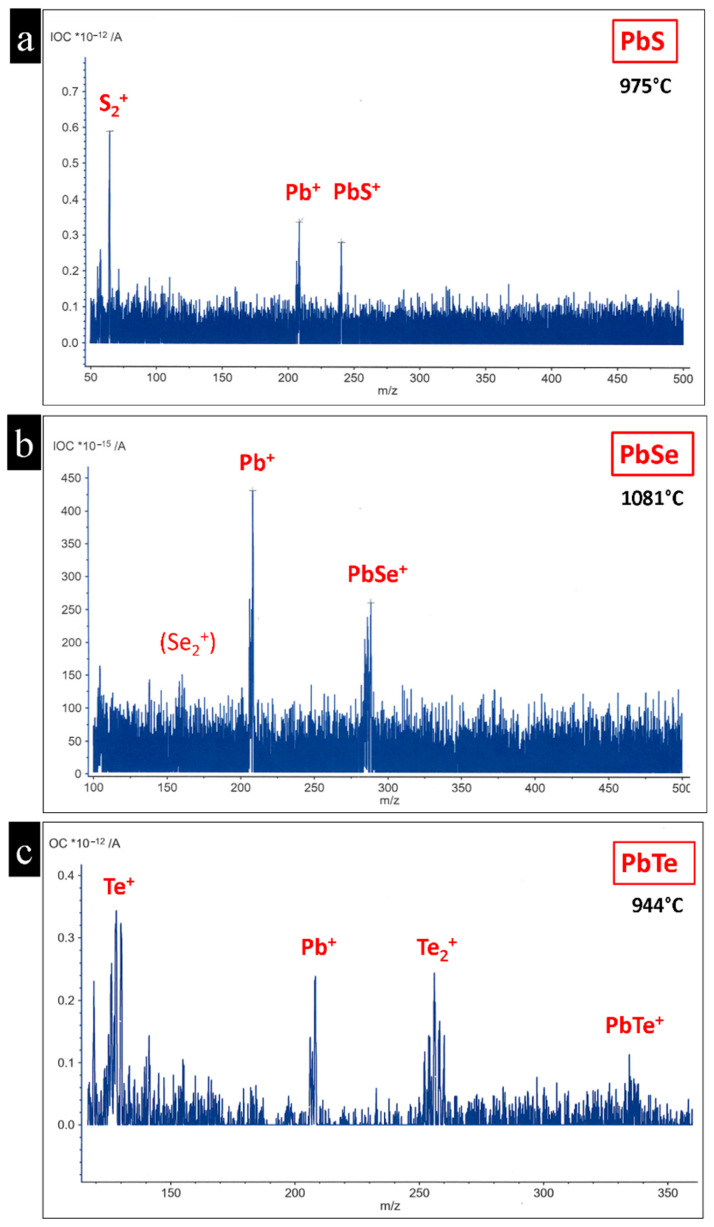
Mass spectra of PbS, PbSe, and PbTe chalcogenides. (**a**) The PbS spectrum reveals atomic Pb and molecular S_2_^+^ and PbS^+^ fragments. (**b**) In PbSe, only two species are identified, atomic Pb^+^ and molecular PbSe^+^ with weak indication for the presence of Se_2_^+^. (**c**) The spectrum of PbTe shows that in addition Te and Pb ions in the gas phase, even heavier molecular ions, such as Te_2_^+^ and PbTe^+^, were also detected.

**Figure 13 nanomaterials-15-01275-f013:**
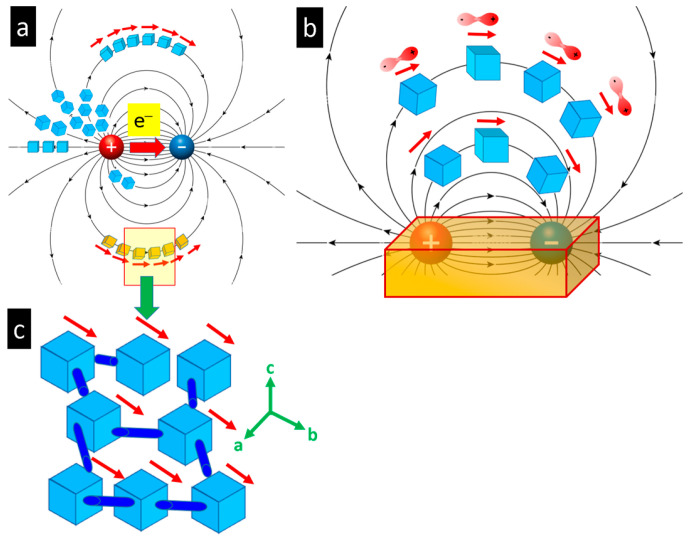
(**a**,**b**) Scenario illustrating synchronized orientation of nanoparticles during in situ formation under electron beam irradiation. The electron flux passing through the sample induces an electric dipole field. The produced nanoparticles are influenced by the generated dipole field in the sample, which ensures common orientation/alignment along the electric field lines. (**c**) Common orientation of nanoparticles in a local segment along the field lines. The atomic interconnections may maintain the relative orientation between the nanoparticles during the growth process, even though heavy dynamical processes may occur, such as thermodiffusion on the substrate.

**Figure 14 nanomaterials-15-01275-f014:**
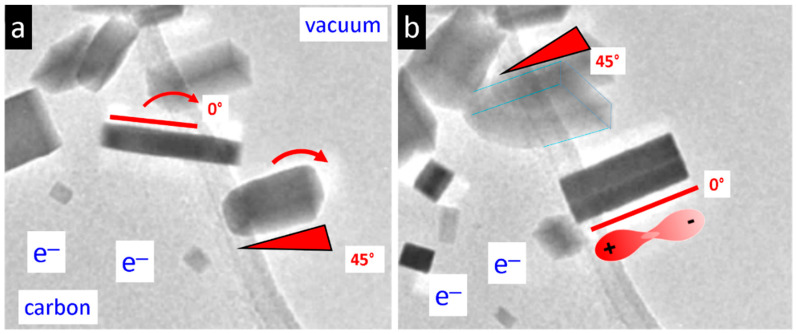
Orientational changes of the nanoparticles during their growth process. Two particles are shown undergoing a tilt angle change of about 45° with respect to the negatively charged carbon substrate. (**a**) At the initial state, the rod at bottom shows an angle of about 45° to the substrate. (**b**) After several seconds of growth, the rod adopts a tilt angle of about 0°, which results in a larger contact area of the positively charged end to the negatively charged carbon.

**Table 1 nanomaterials-15-01275-t001:** Detected gas species of lead chalcogenides measured using mass spectrometry on the single-crystal samples.

Compound	Gas Ion (*m*/*z*)	Gas Ion/(*m*/*z*)	Gas Ion/(*m*/*z*)	Gas Ion/(*m*/*z*)
PbS	-	S_2_^+^ 64	Pb^+^ 208	PbS^+^ 240
PbSe	-	Se_2_^+^ 160	Pb^+^ 208	PbSe^+^ 288
PbTe	Te^+^ 130	Te_2_^+^ 256	Pb^+^ 208	PbTe^+^ 336

**Table 2 nanomaterials-15-01275-t002:** Decomposition temperatures of lead chalcogenides in argon atmosphere, measured using mass spectroscopy. The onset of sublimation was determined by selecting the temperature where Pb^+^ was detectable in the gas phase.

Compound	Sublimation Temp.
PbS	970 °C ± 5 °C (Pb^+^)
PbSe	960 °C ± 5 °C (Pb^+^)
PbTe	925 °C ± 5 °C (Pb^+^)

## Data Availability

The original contributions presented in this study are included in the article/Appendix A. Further inquiries can be directed to the corresponding author.

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
