# Peer review of "In Situ TEM Observation of Electric Field-Directed Self-Assembly of PbS and PbSe Nanoparticles"

_nanomaterials, 2025, doi:10.3390/nano15161275_

Round 1

Reviewer 1 Report

Comments and Suggestions for Authors

Since the self assembly happens in the TEM chamber, the authors are suggested to provide the time dependent TEM and HRTEM images to show how the nanoparticles nucleate and grow. And also the composition with exact atomic ratio should be provided to confirm the formation of PbS, or PbSe.

Author Response

Since the self assembly happens in the TEM chamber, the authors are suggested to provide the time dependent TEM and HRTEM images to show how the nanoparticles nucleate and grow. And also the composition with exact atomic ratio should be provided to confirm the formation of PbS, or PbSe.

Answer:

Time resolved images were already shown in the old version, see actual Figures 7e-h for the nanoparticles growth of PbSe and Figure 6 for the PbSe thin film where the grain pattern transformation during electron beam irradiation is documented. Additionally, we added five videos, three of them showing macroscopic crystals during sublimation PbS_1, PbSe_1, PbSe_2 and two videos of nanoparticles growth and morphogenesis (PbS_2) and thin film nanograin changing during sublimation (PbSe_3). We added an EDX spectrum as Supplementary Figure 1 for the determination of the composition of PbSe.

Reviewer 2 Report

Comments and Suggestions for Authors

The authors need to improve in the following points:

1.The crystal planes isnot presented in corresponding figs;

2.The microscale meter images is not presented.

3. What are self assembly mechanism?

4.How the self assembly are used in relating application?

5. What are the main merits of PbS/PbTe self assembly over conventional film composited by PbS/PbTe such as cvd, pecvd, surface modification or oxidation routes?

6.Why there is no theoretical calculation and verification in terms of nanosized effect?

These are all to be addressed  physiochemically and reasonably.

Comments on the Quality of English Language

minor improvements.

Author Response

1.The crystal planes is not presented in corresponding figs;

2.The microscale meter images is not presented.

  1. What are self assembly mechanism?

4.How the self assembly are used in relating application?

  1. What are the main merits of PbS/PbTe self assembly over conventional film composited by PbS/PbTe such as cvd, pecvd, surface modification or oxidation routes?

6.Why there is no theoretical calculation and verification in terms of nanosized effect?

These are all to be addressed physiochemically and reasonably.

Comments on the Quality of English Language minor improvements.

Answer:

  1. We added in the new high-resolution images Figure 1-3 of the single crystals of PbS and PbSe the coordinates of lattice orientation as well as indicated the corresponding zone and labelled in the FFTs and electron diffraction pattern the main reflections. The new high-resolution images of the nanoparticles are also labeled with coordinates, and zone axes, see Figures 8 and 11 and the corresponding FFT images. In the high-resolution images Figures 9 and 10 we resigned of labeling since in this case, the nanoparticles interconnections should be discussed and emphasized.

  1. The microscale meter images are now shown as follows for the single crystal: light microscopy with new Figures 1a,b of PbSe and Figure 2a,b for PbSe. We inserted overview TEM images as microscale images, see Figures 1c,e for PbS and Figures 2c,d of PbSe. Further on, we show the single crystal particles as whole in Figure 4 and new Figures 11ab, 13 and 14.  

Nanoparticles distribution in microscale overview was already shown in images of old version Figure 1b,d and 2a,b,c and Figure 3. Additionally, we added the microscale meter images for the nanoparticles 11a-c and 14.

  1. Self-assembly is driven by independent condensation of nanoparticles on the carbon substrate emerging from the sublimed material of the mother crystal. During electron beam irradiation of the mother crystal more and more material in form of Pb and Se or S ions/atoms or e.g. Se2 and PbS molecules is supplied leading to further growth of the initial nanoparticles. After sufficient growth period, the particles intergrow to form a few nanometers thick thin film. Further irradiation is leading to OSTWALD ripening, thus large grains integrate the smaller ones giving rise to a more homogenous appearance, see g. time resolved experiment of PbSe in Figure 6 and corresponding new video PbSe_3 in the supplementary data.

As new observation, we added the occurrence of synchronous oriented nanoparticles on the micrometer scale which governs the self-assembly procedure. Here, we assume that possibly an external electric field is monitoring the homogenous crystallographic alignment of the nanoparticles. This can be explained by the presence of electric dipole fields of the nanoparticles at elevated temperatures. The occurrence of dipoles in the PbTe unit cells was calculated and shown in a Science paper in 2010 by Bozin et al., see new citation 38.  

Further on, we observe overall interconnections between the nanoparticles, which keep them certainly in a fixed position with respect to each other during their growth. Similar atomic sized interconnections of PbSe nanoparticles we observed in a macroscopic mesocrystal, however, there the PbSe bridges were created by mineralizing the oleic acid tenside surfactant molecules, see citation 21, Simon et al. 2014.

  1. Here, one could think of fabrication of homogenous thin films which could be prepared without any usage of stabilizers or surfactants and used as infrared sensor e.g. in case of PbSe.

  1. Assumably, electron beam deposition could deliver high quality homogenous thick and structured coatings.

In fact, electron assisted deposition technique is a wide spread technique in industry. The material is sublimed and condensates on different substrates. Herewith, very high coating rate can be achieved, ten to hundred times higher than e.g. by magnetron-sputtering. For example, electron beam sputtering is used for large area coating of UV-light blocking glass in windows. Besides this, electron beam with high power can melt metals with high melting temperatures and thus are used for purification purposes. For industrial application see e.g. https://vonardenne.de /technologien/elektronenstrahltechnologie/. Already in the 1980th, the company von Ardenne installed large fabrication of electron beam assisted deposition and more than 400 devices are still in function all over the world.

  1. The theoretical calculation and verification in terms of nanosized effect we did not perform since there are numerous/myriads of publications dealing on the nanosized chalcogenides and thermoelectrics. For example:

Size Effect on the Structural and Electronic Properties of Lead Telluride Clusters, Y. Mulugeta, H. Woldeghebriel, Int. J. Quant. Chem. 2015, 115, 197–207

Thermoelectric figure of merit of a one-dimensional conductor, I.D. Hicks, M. S. Dresselhaus, Phys. Rev. B, 47, 1993, 16631.

Dynamics and Removal Pathway of Edge Dislocations in Imperfectly Attached PbTe Nanocrystal Pairs: Toward Design Rules for Oriented Attachment. Ondry, J. C., et al. ACS Nano, vol. 12, no. 4, Feb. 2018. https://doi.org/10.1021/acsnano.8b00638.

On the other hand, in case of dislocation dynamics we performed calculations and could explain the transition from metallic to semiconductor behavior for e.g. PbTe upon heating by the rearrangement of the dangling bonds, see citation 34.

Reviewer 3 Report

Comments and Suggestions for Authors

The paper is written in the form which is suitable for Nanomaterials. I appreciate application in-situ TEM method for evaluation of PbS and PbSe semiconductors.

Author Response

The paper is written in the form which is suitable for Nanomaterials. I appreciate application in-situ TEM method for evaluation of PbS and PbSe semiconductors.

Answer:

We thank kindly the referee for his positive evaluation!

Reviewer 4 Report

Comments and Suggestions for Authors

The manuscript entitled “In-situ TEM observation of morphology of nanofilms self-assembly for PbS and PbSe nanoparticles” studied the morphology and phase structure of PbS and PbSe membranes formed by employing in-situ heating process in TEM chamber. The formation mechanisms of the different membranes have also been discussed when combined with the mass spectroscopy method. I recommend to publish the manuscript to Nanomaterials after MAJOR REVISION. The following comments are listed for authors references:

  1. How to precisely confirm the total amount of the evaporated material as 25% of the original particle?
  2. The shape and size of the PbS particles formed after the electron beam irradiation should be confirmed through HRTEM analysis.
  3. Also, the orientation of the irradiated PbSe nanoparticles also need HRTEM or SAED characterization. Bright Field image cannot precisely determine the observation direction.
  4. Obviously, the formed PbSe nanoparticles after irradiation is much more compact than that of PbS nanoparticles when seeing from Figure 2. Please explain the phenomenon.
  5. The authors mentioned that the fast growth of {111} facets lead to the formation of rectangle or square shaped particles of PbSe during the irradiation process. As PbSe possesses FCC structure, the closed packed plane {111} normally exhibit the lowest growth speed, which normally grown with octahedral morphology.
  6. The TEM analysis results do not involved the irradiation process on PbTe, while the discussion part made the conclusion that the PbTe, PbSe and PbS demonstrate similar behavior under the convergent electron beam?
  7. The formation mechanism of the nanopatterns should be discussed in detail.
  8. Is Figure 6c the HRTEM image? If yes, the spots in the figure do not represent the atoms.

Author Response

The manuscript entitled “In-situ TEM observation of morphology of nanofilms self-assembly for PbS and PbSe nanoparticles” studied the morphology and phase structure of PbS and PbSe membranes formed by employing in-situ heating process in TEM chamber. The formation mechanisms of the different membranes have also been discussed when combined with the mass spectroscopy method. I recommend to publish the manuscript to Nanomaterials after MAJOR REVISION. The following comments are listed for authors references:

  1. How to precisely confirm the total amount of the evaporated material as 25% of the original particle?
  2. The shape and size of the PbS particles formed after the electron beam irradiation should be confirmed through HRTEM analysis.
  3. Also, the orientation of the irradiated PbSe nanoparticles also need HRTEM or SAED characterization. Bright Field image cannot precisely determine the observation direction.
  4. Obviously, the formed PbSe nanoparticles after irradiation is much more compact than that of PbS nanoparticles when seeing from Figure 2. Please explain the phenomenon.
  5. The authors mentioned that the fast growth of {111} facets lead to the formation of rectangle or square shaped particles of PbSe during the irradiation process. As PbSe possesses FCC structure, the closed packed plane {111} normally exhibit the lowest growth speed, which normally grown with octahedral morphology.
  6. The TEM analysis results do not involved the irradiation process on PbTe, while the discussion part made the conclusion that the PbTe, PbSe and PbS demonstrate similar behavior under the convergent electron beam?
  7. The formation mechanism of the nanopatterns should be discussed in detail.
  8. Is Figure 6c the HRTEM image? If yes, the spots in the figure do not represent the atoms.

Answer:

  1. The lost of material is hardly to be measured precisely, since the crystal is a 3D object. However, in the TEM image we observe only the 2D projection. Thus, 25 % mass loss is just a rough assessment based on the lateral dimension loss when comparing the crystal before and after irradiation, see Figure 4a and 4b in the new version or Figure 1 in the old version. Thus, we added the remark on page 8 “in this experiment” to exclude the misunderstanding that 25% loss is a general phenomenon. Here, we emphasize that it is only valid for this one specific irradiation experiment shown.

  1. Three high-resolution figures for PbS and PbSe nanoparticles (Figures 8-10). One new overview figure shows common orientation of PbSe nano particles (Figure 11).

  1. Analysis of the before mention three high-resolution figures show that normally the nanoparticles are oriented in 001 [zone] with a rectangular shape and range in the initial state between 2-10 nm in diameter/size, see e.g. Figure 8.

  1. The reason why PbSe the nanoparticles after irradiation are much more compact could be based on the fact that PbSe shows a lower sublimation temperature compared to PbS. Thus, under same conditions with PbS more material of PbSe is condensed on the substrate leading to massive coating. Besides the sublimation temperature, also other thermodynamic data go in this line comparing e.g. crystal sublimation and cohesive energies, lattice energies, enthalpies of formation etc. which are listed in the supplementary.

  1. Yes, the formulation is wrong. We corrected the sentence and changed “fast” into “slow” on page 13, bottom paragraph. Concerning {111} or {110} facets, we observe quite often that the corners of the nanocubes or nanorods remain unfinished, see e.g. Figure 7a,c and h in overview and new high-resolution images of Figures 7a,d and Figure 10c,d. Since the layers along (111) or (110) are densely occupied with atoms compared to the (100) basic plane, it seems that growth along this direction needs more time and thus is more slowly.

  1. PbTe irradiation experiment was mentioned since these results we already published in 2021 and thus we draw similarities to PbS and PbSe, see citation 21. In order to avoid misunderstandings, we removed Figure 6 (of the old version) and the paragraph concerning PbTe.

  1. Here, for me it is unclear whether the nanopattern within the single crystal is meant by the referee or the pattern formation of the condensate nanoparticles.

The nanopattern within the single crystal we show and explain in new Figures 1,2 and 3. There we recognize nanodomains, dislocations, intergrowth etc. A possible model for the nanopattern is given in new Figure 3e. Dislocation formation and dynamics (time and temperature resolved) we already published for PbTe in 2021, see citation 34.

In the case of the nanoparticles, self-assembly is driven by independent condensation of nanoparticles on the carbon substrate emerging from the sublimed material of the mother crystal. During electron beam irradiation of the mother crystal more and more material in form of Pb and Se or S ions/atoms or e.g. Se2 and PbS molecules is supplied leading to further growth of the initial nanoparticles. After sufficient growth period, the particles intergrow to form a few nanometers thick thin film. Further irradiation is leading to OSTWALD ripening, thus large grains integrate the smaller ones giving rise to a more homogenous appearance, see e.g. time resolved experiment of PbSe in Figure 6 and corresponding new video PbSe_3 in the supplementary data. 

As new observation, we added the occurrence of synchronous oriented nanoparticles on the micrometer scale which governs the self-assembly procedure. Here, we assume that possibly an external electric field is monitoring the homogenous crystallographic alignment of the nanoparticles. This can be explained by the presence of electric dipole fields of the nanoparticles at elevated temperatures. The occurrence of dipoles in the PbTe unit cells was calculated and shown in a Science paper in 2010 by Bozin et al., see new citation 38. 

  1. Yes, this image is spherical aberration corrected high-resolution image with 0.66 Ångström resolution. Here, we carried out a time and temperature resolved experiment showing that in case of PbTe single crystal the outermost crystal layer becomes disordered during irradiation giving rise to fibrous atomic sized PbTe chain appearance at the very top surface. The atom per atom removal of the surface layer could be shown. In this case, the outermost chain is disrupted due to the removal of one atom leading to free endings of the chain, see also citation 25, Zelenina et al. in 2021.

Round 2

Reviewer 2 Report

Comments and Suggestions for Authors

Accept

Author Response

Accept

Answer:

We thank kindly the referee for his positive evaluation!